# Gamma-Synuclein Dysfunction Causes Autoantibody Formation in Glaucoma Patients and Dysregulation of Intraocular Pressure in Mice

**DOI:** 10.3390/biomedicines11010060

**Published:** 2022-12-27

**Authors:** Tatiana A. Pavlenko, Andrei Y. Roman, Olga A. Lytkina, Nadezhda E. Pukaeva, Martha W. Everett, Iuliia S. Sukhanova, Vladislav O. Soldatov, Nina G. Davidova, Natalia B. Chesnokova, Ruslan K. Ovchinnikov, Michail S. Kukharsky

**Affiliations:** 1Helmholtz Moscow Research Institute of Eye Diseases, Ministry of Health of the Russian Federation, 105062 Moscow, Russia; 2Institute of Physiologically Active Compounds at Federal Research Center of Problems of Chemical Physics and Medicinal Chemistry, Russian Academy of Sciences, 142432 Chernogolovka, Russia; 3Department of General and Cell Biology, Faculty of Medical Biology, Pirogov Russian National Research Medical University, 117997 Moscow, Russia; 4Department of Pharmacology and Clinical Pharmacology, Belgorod State National Research University, 308015 Belgorod, Russia

**Keywords:** γ-synuclein, autoantibodies, glaucoma, dopamine, α2-macroglobulin, tear fluid, intraocular pressure

## Abstract

Dysregulation of intraocular pressure (IOP) is one of the main risk factors for glaucoma. γ-synuclein is a member of the synuclein family of widely expressed synaptic proteins within the central nervous system that are implicated in certain types of neurodegeneration. γ-synuclein expression and localization changes in the retina and optic nerve of patients with glaucoma. However, the mechanisms by which γ-synuclein could contribute to glaucoma are poorly understood. We assessed the presence of autoantibodies to γ-synuclein in the blood serum of patients with primary open-angle glaucoma (POAG) by immunoblotting. A positive reaction was detected for five out of 25 patients (20%) with POAG. Autoantibodies to γ-synuclein were not detected in a group of patients without glaucoma. We studied the dynamics of IOP in response to IOP regulators in knockout mice (γ-KO) to understand a possible link between γ-synuclein dysfunction and glaucoma-related pathophysiological changes. The most prominent decrease of IOP in γ-KO mice was observed after the instillation of 1% phenylephrine and 10% dopamine. The total protein concentration in tear fluid of γ-KO mice was approximately two times higher than that of wild-type mice, and the activity of neurodegeneration-linked protein α2-macroglobulin was reduced. Therefore, γ-synuclein dysfunction contributes to pathological processes in glaucoma, including dysregulation of IOP.

## 1. Introduction

Glaucoma is a multifactorial disease with various mechanisms of development and is defined as chronic progressive optic neuropathy with morphological changes in the optic nerve head and the layer of retinal nerve fibers associated with the death of retinal ganglion cells and visual field defects [1]. It is the second-leading cause of blindness worldwide and the most common cause of irreversible blindness [2,3]. Pathological processes in glaucoma have common features with neurodegenerative diseases, including the death of a specific nerve cell type—retinal ganglion cells (RGCs). An increase in intraocular pressure (IOP) is the most common cause of the development of this pathology. Drug therapy for glaucoma is mainly aimed at reducing IOP [4]. However, a form of glaucoma occurring with normal IOP (normal-tension glaucoma) emphasizes the complex nature of pathogenesis. Glaucoma development involves processes such as neuroinflammation, impairment of the immune response, disturbance of nerve impulse transmission, and other mechanisms underlying neurodegeneration. Studying them may reveal new possibilities for the prognosis and treatment of glaucoma [5,6,7].

Synucleins are highly conserved cytosolic proteins involved in the regulation of synaptic transmission in the nervous system [8,9,10]. The family consists of three highly homologous proteins: α-, β-, and γ-synuclein. α-synuclein is actively studied because of its role in the pathogenesis of neurodegenerative diseases. α-synuclein is the main component of Lewy bodies in Parkinson’s disease (PD) and dementia with Lewy bodies (DLB). It is also found in senile plaques in Alzheimer’s disease [11,12,13]. Aggregation of γ-synuclein in the nervous system is detected in amyotrophic lateral sclerosis (ALS) and DLB [14,15,16,17]. γ-synuclein is also associated with cancer development [18,19]. All three synucleins are expressed in the retina and the optic nerve, but only γ-synuclein can change the level and distribution of its expression in glaucoma [20]. γ-synuclein could be considered as a specific marker of RGCs because of its high level of expression in these cells. The redistribution of γ-synuclein from one layer of the retina and the optic nerve to another leads to a decrease in its level in the original location. Moreover, the death of RGCs in glaucoma correlates with a decrease in the expression of γ-synuclein [21].

In neurodegenerative diseases, autoantibodies against proteins prone to aggregation are found in the blood [22]. Thus, autoantibodies against α-synuclein are found in patients with PD [23]. We have previously shown that autoantibodies to γ-synuclein are detected in the serum of some patients with ALS and cerebrovascular diseases [24]. The exact role of autoantibodies in the pathogenesis of ocular diseases remains unclear. They can cause autoimmune death of retinal cells. They can also have a protective effect [25,26]. In particular, autoantibodies against γ-synuclein increase the survival of RGCs in RGC-5 cell culture under stress conditions and in primary cultures of porcine retinal explant [27,28]. Therefore, despite the evidence of γ-synuclein involvement in glaucoma development, the particular mechanisms remain unclear.

We found that some patients with primary open-angle glaucoma (POAG) had autoantibodies to γ-synuclein in their blood. Using knockout mice (γ-KO mice) as a model for the loss of function of this protein, we showed that the impact on the neurotransmitter systems of the eye responsible for IOP regulation differently affects the dynamics of IOP changes in γ-KO mice compared to wild-type mice (WT mice). In the tear fluid of γ-KO mice, the activity of chaperone and immune system modulator α2-macroglobulin (α2-MG) was reduced.

## 2. Materials and Methods

### 2.1. Patients

The study involved 25 patients with POAG, including 12 women and 13 men. The control group of apparently healthy individuals without a diagnosis or history of glaucoma included 13 people (7 women and 6 men) (Table 1). Patients were followed at the Helmholtz Moscow Research Institute of Eye Diseases of the Ministry of Health of the Russian Federation. All patients underwent a comprehensive ophthalmological examination: viscometry, pneumotonometry, biomicroscopy, gonioscopy, ophthalmoscopy, and static perimetry. All glaucoma patients received instillation of ocular hypotensive drugs to normalize IOP. The early stage of the disease was diagnosed in 5 (20%) cases; a moderate stage was diagnosed in 7 (28%) cases; an advanced stage was diagnosed in 8 (32%) cases; a severe stage was diagnosed in 4 (16%) cases; and end-stage glaucoma was diagnosed in 1 (4%) case (Table 1). The mean age of patients and volunteers without glaucoma was 73 ± 8 and 66 ± 12 years, respectively. All patients provided written informed consent at admission. The study was conducted according to the guidelines of the Declaration of Helsinki, and approved by the Institutional Ethics Committee of the HMRIED (protocol No. 55/3, 17 June 2021).

The exclusion criteria were significant refractive errors (high myopia and hyperopia, astigmatism above 2.0 diopters), non-glaucoma pathology of the optic nerve, severe clouding of the cornea, and a history of acute cerebrovascular accidents. Blood taken for the standard biochemical analysis was used for the study. The serum was obtained and stored at a temperature of −80 °C.

### 2.2. Detection of Autoantibodies to γ-Synuclein

Immunoblotting was used to detect autoantibodies to γ-synuclein in blood serum [24]. Electrophoretic separation of 0.15 μg of purified recombinant human γ-synuclein protein in 14% sodium dodecyl sulfate–denaturing polyacrylamide gel was performed. Then, the samples were transferred to a polyvinylidene fluoride (PVDF) membrane (Hybon-P, Amersham, Sheffield, UK) using a semi-dry method. Membrane block was performed in a solution of 4% non-fat dry milk (NFDM) prepared in saline buffer TBST (Tris-buffered saline, 0.1% Tween 20). Then, it was incubated with the serum samples diluted 200 times in a solution of 4% NFDM in TBST overnight at 4 °C. Further incubation was performed with antibodies conjugatedwith horseradish peroxidase against human IgG (Bio-Rad, Hercules, CA, USA) in the same buffer as the serum samples (1:3000 dilution) for 1.5 h at room temperature. The detection of the resulting complex of primary and secondary autoantibodies was performed using a chemiluminescence detection kit (ECL Plus, Thermo Fisher Scientific, Waltham, MA, USA) according to the manufacturer’s instructions. X-ray film (Thermo Scientific, Waltham, MA, USA) was used to detect chemiluminescence. A band corresponding to the molecular weight of γ-synuclein 17 kDa was detected on the X-ray film if the antibodies to γ-synuclein were present in the tested serum.

### 2.3. Experimental Animals

We used γ-synuclein (Sncg) knockout mice described previously [29]. The animals were provided by Professor V. Buchman. The name and catalog number of The Jackson Laboratory are B6.129P2-Sncgtm1Vlb/J and 008843, respectively. Homozygous γ-KO mice and WT control mice were generated by intercrossing heterozygous knockout and C57BL/6J mice. Experimental animals were housed at a 12 h light/12 h dark cycle, with food and water supplied ad libitum. The procedures were carried out in accordance with the “Guidelines for accommodation and care of animals. Species-specific provisions for laboratory rodents and rabbits” (GOST 33216-2014) in compliance with the principles enunciated in Directive 2010/63/EU on the protection of animals used for scientific purposes and approved by the local Institute Ethics Review Committee of the IPAC RAS (protocol No. 48, 15 January 2021). All efforts were made to minimize the number of animals and their suffering. All mice were genotyped using PCR analysis of DNA obtained from an ear biopsy, as described elsewhere [30].

Experimental animals (γ-KO and WT mice) underwent single instillations of 10 μL of 1% phenylephrine hydrochloride (Mesaton, Dalkhimfarm, Khabarovsk, Russia), 0.5% timolol maleate (Timolol, Diapharm, Moscow, Russia), 1% pilocarpine (Pilocarpine, Renewal, Novosibirsk, Russia), 0.1% atropine sulfate (Atropine, Dalkhimfarm, Khabarovsk, Russia), 10% dopamine (Sigma-Aldrich, Burlington, MA, USA) in 0.9% NaCl, and 0.25% haloperidol (Haloperidol, Velpharm, Moscow, Russia) in both eyes. IOP was measured under general anesthesia (Avertin, 200 mg/kg, intraperitoneally) using the automatic electronic tonometer Tonovet (Icare Finland Oy, Vantaa, Finland) in the morning, before and after instillation (1 and 2 h later). In our preliminary experiments, it was established that general anesthesia by Avertin does not affect the level of IOP in either γ-KO or WT mice.

Tear fluid was collected from both eyes with a filter paper strip (2.5 mm width wide), which was placed behind the lower eyelid for 5 min. Then, tear fluid components from both eyes from a mouse were eluted for 20 min with saline (50 μL) in one tube, centrifuged for 10 min at 3000 rpm, and the supernatant was used for testing. In the tear fluid eluate, the protein concentration was determined according to Lowry [31], and the activity of α2-MG was determined by the enzymatic method with the specific substrate N-benzoyl-DL-arginine-p-nitroanilide (BAPN), as described previously [32]. The activity of α2-MG was expressed as nmol/min × mL of tear fluid and nmol/min × mg of protein. The optical density of the samples was determined on a multifunctional photometer for microplates Synergy MX (BioTek, Winooski, VT, USA). Statistical analysis was performed using GraphPad Prism 6 software (GraphPad Software, San Diego, CA, USA).

## 3. Results

### 3.1. Autoantibodies to γ-Synuclein Were Found in the Blood Serum of Some Patients with Glaucoma

γ-synuclein autoantibodies were detected in the blood serum of 20% of the patients (5 out of 25) with POAG (Figure 1a,b and Appendix A and Table 1). In the control group, γ-synuclein autoantibodies were not detected in the blood serum.

All samples with a positive reaction were reanalyzed to confirm the results. In most cases, immunoblotting showed a single immunoreactive band of 17 kDa, corresponding to γ-synuclein. However, in one case (sample 38), two bands were observed, which may be due to partial degradation of the recombinant protein or contamination of the sample with bacterial proteins. The signal intensity varied for different patients. The high band intensity for some patients (sample 26 in Figure 1a) may indicate a fairly high titer of γ-synuclein autoantibodies in this patient.

Thus, the presence of γ-synuclein autoantibodies in the blood serum was detected in 20% of the patients with POAG. Patients with detected autoantibodies to γ-synuclein were diagnosed with different stages of glaucoma development from 1a to 4a. No correlation was found between the stage of the pathological process, the IOP level, and the presence of γ-synuclein autoantibodies. Subsequent studies on a greater number of patients may be required to obtain more reliable results.

### 3.2. Loss of γ-Synuclein Function Led to Dysregulation of IOP in Mice

We studied IOP changes in γ-KO mice during the instillation of drugs affecting IOP regulation to determine what effect lack of γ-synuclein can have on hydrodynamics of the eye.

We used agonists and antagonists for the three main mediator systems that control IOP: adrenergic agonist phenylephrine, adrenergic antagonist timolol, cholinergic agonist pilocarpine, cholinergic antagonist atropine, dopamine agonist dopamine, and dopamine antagonist haloperidol. There was a difference in the dynamics of changes in IOP between the group of γ-KO mice and WT mice after instillation into the eyes of mice with drugs that affect one of the three mediator systems.

The adrenergic agonist 1% phenylephrine caused IOP reduction in γ-KO mice by an average of 4 mmHg after 1 h (*p* = 0.0017) and 3 mmHg after 2 h (*p* = 0.0222). In WT mice, there was a tendency to the opposite dynamic of changes in IOP. However, there were no significant differences between the first (0 h) and subsequent (1 h and 2 h) time points (Figure 2a). The adrenergic antagonist 0.5% timolol caused IOP reduction in both groups. However, a significant difference between time points was achieved only for WT mice, with no differences between groups (Figure 2b).

The cholinergic agonist 1% pilocarpine did not cause any changes in IOP dynamics relative to baseline values and between groups (Figure 2c). The cholinergic antagonist 0.1% atropine caused a significant IOP reduction in γ-KO mice after 2 h by an average of 2 mmHg (*p* = 0.0271), compared to that of WT mice, as well as in comparison with the initial value by 3 mmHg (0.0018). In WT mice, atropine did not affect the level of IOP (Figure 2d).

The dopamine agonist 10% dopamine caused IOP reduction in γ-KO mice from 8 to 4 mmHg (*p* = 0.0011) after 1 h and from 8 to 5 mm Hg. (*p* = 0.0011) after 2 h from baseline. IOP in WT mice was twice as high as IOP in γ-KO mice throughout the experiment (*p* = 0.0004 and *p* < 0.0001 for 1 h and 2 h, respectively). IOP in WT mice did not change significantly (Figure 2e). The dopamine antagonist 0.25% haloperidol caused IOP reduction after 1 h in both groups, relative to the initial values. After 2 h, the IOP level in WT mice returned to the original values, while in γ-KO mice it remained lower by 2 mmHg than that of the WT mice (*p* = 0.0063, Figure 2f).

Therefore, the impact on any of the three mediator systems, at least in one direction (stimulation or inhibition), led to changes in the IOP dynamic in γ-KO mice compared to WT mice. The most prominent effect was manifested upon stimulation of dopamine reception, and IOP changes in knockout mice were observed with the use of both an agonist and an antagonist of dopamine receptors.

### 3.3. The Total Protein Concentration and the Activity of the α2-Macroglobulin Changed in the Tear Fluid of γ-KO Mice

In the tear fluid of γ-KO mice, a higher content of total protein was found than in the tear fluid of WT mice (Figure 3a). Measuring α2-MG activity in the tear fluid of γ-KO mice showed a tendency to the activity decrease normalized to the volume of tear fluid (Figure 3b), and a significant decrease by 70% normalized to the total amount of protein (mg) (Figure 3c).

## 4. Discussion

Autoantibodies to γ-synuclein are not detected in individuals without glaucoma. The presence of autoantibodies to γ-synuclein in the blood of 20% of the patients with POAG indicates that this protein, at least in some cases, is involved in the pathogenesis of glaucoma. Based on the multifactorial nature of glaucoma development, we assume that γ-synuclein is involved in this process in varying degrees. Previously, we showed that autoantibodies against γ-synuclein are also detected in patients with chronic cerebral ischemia, with the frequency of detection of one in 7.6 people, and in patients with ALS, with the frequency of detection of one in 20.5 people [24]. The results of this study indicate that this rate is higher in patients with glaucoma (one in five people). We assume that a common mechanism explaining the formation of autoantibodies to γ-synuclein in the above-mentioned diseases is γ-synuclein or its oligomeric-aggregated forms entering the bloodstream in cases of histohematic barriers disruption. Some proteins, including synucleins, change localization and accumulate in certain layers and/or types of retinal cells in neurodegenerative diseases [33,34]. Moreover, in neurodegeneration, the aggregation of protein occurs in the retina, as it does in the brain [35,36]. In glaucoma, changes in the expression and localization of γ-synuclein were noted in the optic nerve. Optic nerve fibers are normally immunopositive for γ-synuclein. However, strong γ-synuclein immunopositive staining of the optic nerve was observed in the area of lamina cribrosa and in the postlamina area only in eyes with glaucoma. A model of glaucoma reproduced by cauterization of episcleral veins shows a decrease in the amount of both γ-synuclein mRNA and the protein in the optic nerve. During the incubation of rat astrocyte culture at elevated hydrostatic pressure, the amount of γ-synuclein decreased [20]. The expression of γ-synuclein at the mRNA and protein levels decreased in the retina of genetic glaucoma (DBA/2J) mice and optic nerve crush (ONC) mice [37,38]. At the same time, accumulation of γ-synuclein was observed only in the optic nerve of ONC mice but not in that of DBA/2J mice [37]. Despite the obtained experimental evidence of changes in the expression and localization of γ-synuclein in the eye during the glaucomatous process, the exact mechanisms of implication of this protein in the pathogenesis of glaucoma remain unclear. The formation of autoantibodies against γ-synuclein and the lack of protein function may contribute to the pathogenesis of glaucoma. Nevertheless, some data indicate that autoantibodies against γ-synuclein can be a protective factor that ensures removing γ-synuclein excess during a critical increase in its intracellular concentration and removing its misfolded and aggregated forms. This is indirectly confirmed by the neuroprotective effect of autoantibodies to γ-synuclein in cultured neuroretinal cells [27,28].

One of the mechanisms of γ-synuclein involvement in glaucoma pathogenesis may be its participation in the neurotransmission of monoamines, which may affect IOP [39,40]. All three members of the synuclein family modulate the transport, expression, and function of monoamine transporters on the cell surface, thereby playing a central role in the regulation of monoamine reuptake [40]. In γ-synuclein knockout mice, behavioral testing revealed a moderate deficiency of the dopaminergic neurotransmitter system [41]. γ-synuclein can regulate the activity of the serotonin transporter that reuptakes serotonin from the synaptic cleft [42]. The level of IOP depends on the rate of outflow and the formation of intraocular fluid, which is influenced by the adrenergic, dopaminergic, and cholinergic systems [39]. In this study, we showed that the IOP dynamic response is different in γ-KO mice compared to that in WT mice when mediator regulators are instilled. IOP decreases in γ-KO mice after instillations of a dopaminergic agonist and antagonist, an adrenergic agonist, and an antagonist of cholinergic receptors. At the same time, IOP does not change or tends to increase in WT mice, as in the case of phenylephrine. This may be due to dysfunction of the neuronal control of IOP in γ-KO mice and disruption of the correct interaction among mediators of the adrenergic, dopaminergic, and cholinergic systems [43,44]. Further investigation is needed with additional selective receptor agonists and antagonists.

One of the markers of a neurodegenerative process, not only in the retina but also in the brain is an increase in the level of α2-MG in the tear fluid [32]. α2-MG is a protein of acute phase inflammation, an inhibitor of a wide range of proteolytic enzymes, and a regulator of the function of many cytokines and growth factors. As an extracellular chaperone, α2-MG prevents the formation of misfolded proteins and their conglomerates, including amyloid-β, which accumulates in the retina in glaucoma [45,46,47,48,49]. We noticed a significant increase in the concentration of total protein in the tear fluid of γ-KO mice. This is possibly related to a decrease in the water flow into the tear fluid from the lacrimal glands and the formation of a more concentrated tear fluid. A significant decrease of α2-MG activity in the tear fluid of γ-KO mice indicates a dysregulation of its protective effect. An increase in α2-MG activity is a response to neuroinflammation and the formation of malformed proteins, aimed at protecting nerve cells from death [46]. Therefore, our data and other data indicate that the lack of γ-synuclein leads to weakening of protective mechanisms that prevent the development of neurodegenerative processes, including such processes in the eyes.

## 5. Conclusions

The detected autoantibodies in 20% of the patients with POAG confirmed the involvement of this factor in the pathogenesis in some forms of glaucoma. The change in IOP response to the instillation of drugs that affect the hydrodynamics of the eye in γ-KO mice indicated that the loss of this protein function can lead to IOP dysregulation. In the tear fluid of γ-KO mice, the activity of α2-MG controlling the processes of neuroinflammation and the formation of misfolded proteins was reduced, whereas the total amount of protein was increased. Therefore, γ-synuclein dysfunction in glaucoma contributes to impaired IOP regulation and development of neurodegenerative processes in the retina—the main factors leading to optic neuropathy and loss of visual functions in glaucoma.

## Figures and Tables

**Figure 1 biomedicines-11-00060-f001:**
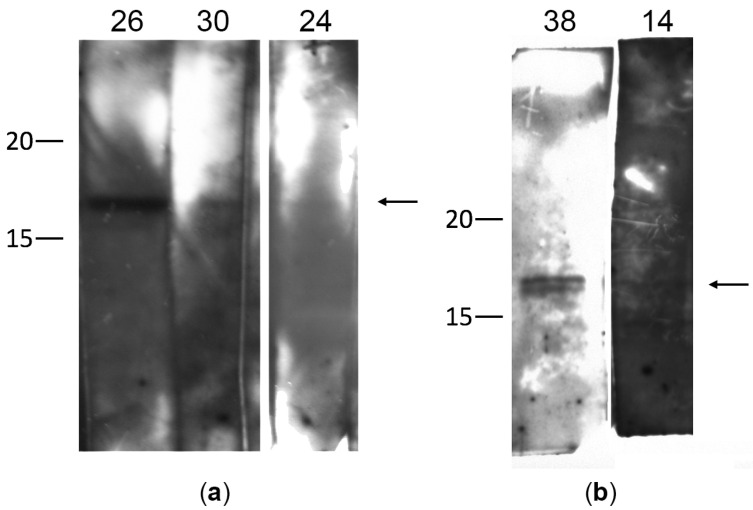
Detection of autoantibodies to γ-synuclein in the serum of patients with glaucoma by immunoblotting: (**a**,**b**) correspond to different independent membranes divided into strips. Each strip of membrane was incubated with a serum sample from one patient. On the left, the molecular weight (kDa) is indicated according to the protein ladder. The numbers 14, 24, 26, 30, 38 indicate the numbers of serum samples from patients with glaucoma, in which autoantibodies to γ-synuclein (17 kDa) were detected. The arrow on the right marks the band corresponding to γ-synuclein.

**Figure 2 biomedicines-11-00060-f002:**
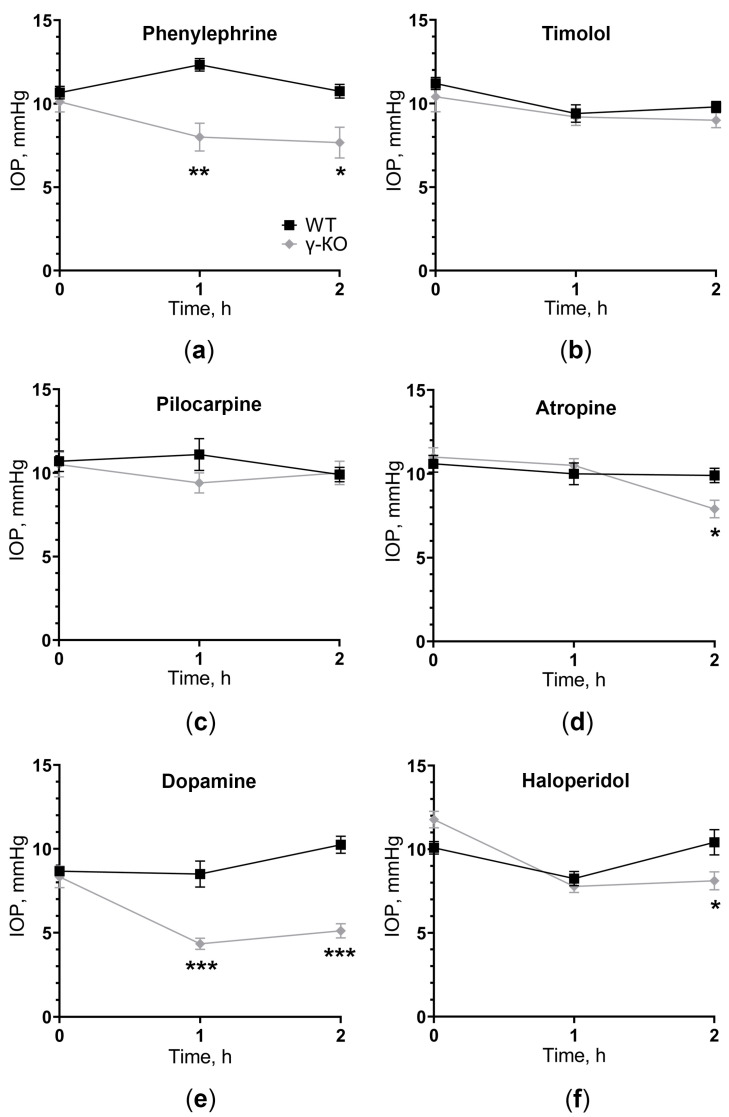
IOP change in γ-synuclein knockout (γ-KO) mice compared to wild-type (WT) mice after instillation of drugs affecting different neuromediators: (**a**) adrenergic agonist 1% phenylephrine, (**b**) adrenergic antagonist 0.5% timolol, (**c**) cholinergic agonist 1% pilocarpine, (**d**) cholinergic antagonist 0.1% atropine, (**e**) dopamine agonist 10% dopamine and (**f**) dopamine antagonist 0.25% haloperidol. IOP was measured before (0 h) and 1 and 2 h after drugs installation. Mean ± SE are presented. Statistical analysis was performed using ANOVA followed by Holm-Sidak’s multiple comparisons test between groups (* *p*  < 0.05, ** *p*  < 0.05, *** *p*  < 0.001). For each group, *n* = 5 (10 eyes).

**Figure 3 biomedicines-11-00060-f003:**
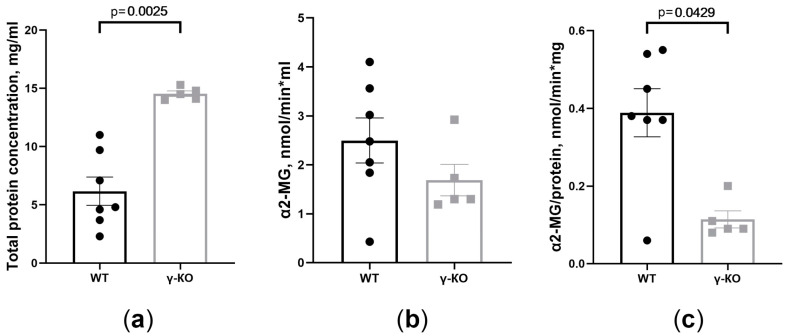
Total protein concentration and α2-macroglobulin (α2-MG) activity are different in tear fluid of γ-synuclein knockout (γ-KO) mice than those of wild-type (WT) mice: (**a**) total protein concentration determined by the Lowry assay (mg/mL); (**b**) total activity (nmol/min × mL); (**c**) specific activity (nmol/min × mg) of α2-MG in tear fluid of γ-KO and WT mice. Means ± SE with individual data are presented and p values are given if differences between groups were statistically significant according to the Mann–Whitney test. For the WT mice group, *n* = 7; for the γ-KO group, *n* = 5.

**Table 1 biomedicines-11-00060-t001:** Clinical data of patients and control subjects.

Patient Number	Gender	Age (Years)	Diagnosis	Autoantibodies to γ-Synuclein	Stage of the Disease	Surgical Treatment of Glaucoma
Control group						
1	M	44	No glaucoma symptoms	-	NA	NA
2	F	91	No glaucoma symptoms	-	NA	NA
3	M	70	No glaucoma symptoms	-	NA	NA
4	M	85	No glaucoma symptoms	-	NA	NA
5	M	64	No glaucoma symptoms	-	NA	NA
6	M	64	No glaucoma symptoms	-	NA	NA
7	F	80	No glaucoma symptoms	-	NA	NA
8	F	59	No glaucoma symptoms	-	NA	NA
9	F	48	No glaucoma symptoms	-	NA	NA
10	F	67	No glaucoma symptoms	-	NA	NA
11	F	63	No glaucoma symptoms	-	NA	NA
12	M	62	No glaucoma symptoms	-	NA	NA
13	F	67	No glaucoma symptoms	-	NA	NA
Primary open-angle glaucoma						
14	M	80	POAG	+	4 OU	Yes
15	F	78	POAG	-	3 OU	No
16	M	76	POAG	-	5 OS; 2 OD	Yes
17	M	78	POAG	-	2OU	Yes
18	F	78	POAG	-	1 OU	No
19	M	78	POAG	-	4 OS; 1 OD	No
20	F	86	POAG	-	1 OU	No
21	F	75	POAG	-	3 OU	Yes
22	M	69	POAG	-	4OS; 3 OD	Yes
23	F	59	POAG	-	1 OS glaucoma suspect; 1 OD	Yes
24	M	76	POAG	+	3 OS; 1 OD	Yes
25	F	69	POAG	-	OU glaucoma suspect	No
26	M	80	POAG	+	2 OS; 1 OD	Yes
27	M	69	POAG	-	1 OS; 3 OD	Yes
28	F	80	POAG	-	2 OU	No
29	M	53	POAG	-	3 OU	Yes
30	M	76	POAG	+	2 OS; 3 OD	Yes
31	F	85	POAG	-	2 OS, OD anophthalmia	No
32	M	64	POAG	-	3 OU	No
33	F	68	POAG	-	OU glaucoma suspect	No
34	F	71	POAG	-	3 OS; 1 OD	No
35	M	79	POAG	-	4 OS; 3 OD	Yes
36	F	68	POAG	-	1 OS; 2 OD	Yes
37	F	71	POAG	-	2 OU	Yes
38	M	65	POAG	+	1 OS; 2 OD	No

“+”—positive for autoantibodies to γ-synuclein; “-”—no autoantibodies to γ-synuclein detected; NA—not applicable.

## Data Availability

Data are contained within the current article and its Appendix A.

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
