# Peer review of "Gamma-Synuclein Dysfunction Causes Autoantibody Formation in Glaucoma Patients and Dysregulation of Intraocular Pressure in Mice"

_biomedicines, 2022, doi:10.3390/biomedicines11010060_

Round 1
Reviewer 1 Report
The manuscript by Pavlenko et al is focused on the unravelling of the role of gamma-synuclein in dysregulation of intraocular pressure and glaucoma. The authors shown that gamma-synuclein can modulate the mechanism of pathology of glaucoma through effect on intraocular pressure via interaction with catecholamines (dopamine). This is interesting and novel manuscript and I have only minor comments.
1. Abstract. "γ-synuclein is an aggregation-prone protein involved in neurodegenerative diseases." However, the role of this form of synuclein in the mechanism of neurodegeneration is disputable compare to alpha-synuclein. Considering this this abstract would be better without this statement.
2. Figure legends 2 and 3 needs some additional explanation.
Reviewer 2 Report
This is a well written and organized manuscript, in which, the authors demonstrated the presence of autoantibodies against γ-synuclein in 20% of the patients in the study.
Specific suggestions/comments raised by the reviewer:
Line 22: “glaucoma-related physiological changes” or “glaucoma-related pathophysiological changes” Please specify.
Line 85: Render “antihypertensive” as “ocular hypotensive”
Line 93: Provide an alternate rendering for “optical cornea”
Lines 134 – 135: It was previously established that general anesthesia does not affect the level of IOP in both γ-KO and WT mice. Previously established by whom and when…
What is the sample size of the experimental animals?
Line 158: Rearrange the order of these numbers.
Line 297: The authors should elaborate on these other forms of glaucoma in which autoantibodies against γ-synuclein are expressed as the manuscript appears to only address POAG.
Line 296: Can you offer any explanation as to why only 1 in 5 individuals with POAG had autoantibodies against γ-synuclein?
